# Desiccation Sensitivity Characteristics and Low-Temperature Storage of Recalcitrant *Quercus variabilis* Seed

**Ming-Jia Zhang** [1,2], **Yong-Zheng Wang** [3], **Yang Xian** [2], **Cheng-Cheng Cui** [2], **Xiao-Man Xie** [2], **Bo-Qiang Tong** [2,*] **and Biao Han** [2,*]

[1] College of Biological Sciences and Technology, Beijing Forestry University, Beijing 100083, China; 2021120336@sdau.edu.cn
[2] Key Laboratory of State Forestry and Grassland Administration Conservation and Utilization of Warm Temperate Zone Forest and Grass Germplasm Resources, Shandong Provincial Center of Forest and Grass Germplasm Resources, Jinan 250102, China; xianyang73@shandong.cn (Y.X.); zhangjihe@sinotruk.com (C.-C.C.); xxm529@shandong.cn (X.-M.X.)
[3] College of Forestry, Shandong Agricultural University, Taian 271000, China; 2022120348@sdau.edu.cn
[*] Correspondence: tbq1001@shandong.cn (B.-Q.T.); hanbiao3361@shandong.cn (B.H.); Tel.: +86-133-5666-2611 (B.-Q.T.); +86-189-5312-3361 (B.H.)

**Abstract:** This study aims to investigate the desiccation sensitivity characteristics and the critical moisture content of the recalcitrant *Quercus variabilis* seed. Additionally, cryopreservation of the recalcitrant seeds were studied. Wild-collected *Q. variabilis* seeds were used in this research. Differential scanning calorimetry (DSC) was employed to evaluate the critical moisture content and germination indices at different moisture contents were measured. The initial moisture content of the seeds and embryonic axes decreased from 33.1% and 40.9%, respectively, to 10.0%, accompanied by a germination rate decrease of 95.6% and 90.0% to 6.3% and 60.0%, respectively. The theoretical critical moisture content of the embryonic axis was calculated to be 11.55%. As dehydration progressed, drastic changes were observed in the antioxidant enzyme system. Initially, the levels of PRO and SOD in the seeds increased and then decreased, while the levels of POD and MDA consistently increased. The cryopreservation of the embryonic axis was achieved using vitrification. The embryonic axis with a moisture content of 10% had a 15% survival rate when pretreated with PVS2 for 60 min prior to cryopreservation. The results demonstrated that the cryopreservation of the *Q. variabilis* embryonic axis is possible, but the method needs adjustment to increase the recovery survival rate.

**Keywords:** *Quercus variabilis*; desiccation sensitivity; cryopreservation; recalcitrant seeds; differential scanning calorimetry (DSC)

## 1. Introduction

Plant seeds can be classified into three categories based on their moisture characteristics during storage: recalcitrant seeds, intermediate seeds, and orthodox seeds [1,2]. Recalcitrant seeds are sensitive to desiccation and low temperatures, and cannot be stored in seed banks like orthodox seeds. The moisture content of seeds has been shown to be the key factor determining their viability and storage life [3]. Most of the moisture in recalcitrant seeds exists as free water, which keeps the seed cells in a high metabolic state [4]. This means that it is difficult for them to tolerate desiccation stress, and recalcitrant seeds are damaged by desiccation when the moisture content drops below a critical level [5]. Differential scanning calorimetry (DSC) is an effective tool for analyzing changes in the thermodynamic properties of moisture within seeds [6]. DSC, combined with indicators such as germination percentage, can therefore predict the critical moisture content of seeds, and can, in turn, be beneficial in analyzing the desiccation sensitivity of recalcitrant seeds [7]. Cryopreservation is currently viewed as the only viable technique for the long-term preservation of recalcitrant seeds. However, recalcitrant seed sensitivity to desiccation

and low temperatures can lead to cryogenic injury during the preservation process [8,9]. Finding an appropriate balance point (critical moisture content) or protective solutions to prevent injuries is a crucial factor in successfully preserving recalcitrant seeds [10,11]. Embryonic axes are small in size and require less space than complete seeds, although they still carry complete genetic information. They can germinate independently and have been demonstrated to be more resistant to desiccation and low temperature than seeds [11]. The cryopreservation of embryonic axes has therefore been proposed as a potential alternative method to preserve plant species with recalcitrant seeds.

*Quercus variabilis* (Fagaceae) is one of the most widely distributed *Quercus* species in China and is the dominant foundation species in deciduous forests in the northern regions of the country. Due to its well-developed root system, *Q. variabilis* is able to withstand drought and poor soil conditions. Additionally, its wood is of high quality, making it an important species for ecological conservation, moisture resource conservation, and timber production. *Q. variabilis* has recalcitrant seeds. Previous studies on the storage of these seeds focused mostly on short-term storage, and due to uneven conditions such as differing storage costs and pre-treatments, no scientific and systematic study into preservation methods has been conducted to date.

DSC has been used to analyze the thermograms of moisture in the seeds of several species, for example *Fortunella polyandra* [6] and *Castanea mollissima* [7], and it is also used to distinguish between *Nicotiana tabacum* seed phenotypes [12]. The potential of the embryonic axis in the cryopreservation of plant species with recalcitrant seed has received some research attention. Wen et al. [13] demonstrated that following the cryopreservation of the embryonic axis for two years, the embryo viability of *Livistona chinensis*, which has recalcitrant seeds, was almost unchanged compared with fresh embryonic axes. The treatment time and concentration of the pre-treatment protective agents were found to have a significant impact on the effectiveness of the cryopreservation of the *Quercus variabilis* embryonic callus [14].

Research on the storage of seeds from forest trees is crucial for cultivating forest trees, forestry production, and conserving germplasm resources. It is also a vital aspect of seed bank construction. However, there is a lack of research on the critical moisture content and cryopreservation of *Q. variabilis* seeds, hindering preservation efforts for this species. As we work towards developing seed banks for preserving forest germplasm resources, it is necessary to explore the desiccation sensitivity of recalcitrant seeds and enhance storage technologies, especially the cryopreservation of recalcitrant seeds. We hope that this research will provide a theoretical reference for research into the preservation of *Q. variabilis* and other species with recalcitrant seeds.

## 2. Materials and Methods

### 2.1. Plant Materials

The *Q. variabilis* seeds were collected from wild individuals growing in the Culai Mountain National Forest Park, China (longitude 117°14′ E, latitude 36°2′ N). All seeds originated from the same half-sib family.

### 2.2. Desiccation Sensitivity of Q. variabilis Seeds (Embryonic Axis)

2.2.1. Determination of Moisture Content

Based on the method to determine seed moisture content described by the International Seed Testing Association (ISTA, 2021), *Q. variabilis* seeds (4 ± 1 g) and embryonic axes (1.5–2.5 g) were selected that were uniform in size and weight. The fresh weight of the complete seeds and embryonic axes was measured, and the seeds and embryonic axes were then dried in an oven at 103 °C for 17 hours (h). The dry weight and the moisture content were then calculated. We used a Mettler-Toledo XSE205high-precision analytical balance

(maximum weighing weight 81 g, accuracy 0.01 mg), with three repetitions using ten seeds per set. The formula used for the calculation of moisture content is:

Moisture content = (Fresh weight − Dry weight)/Fresh weight × 100%

2.2.2. Preparation of Seeds (Embryonic Axes) with Different Moisture Contents

*Q. variabilis* seeds and embryonic axes with a moisture content of 30%, 25%, 20%, 15%, or 10% were prepared using a silica gel weight loss method. Briefly, *Q. variabilis* seeds of the same size and weight (4 ± 1 g) were placed in a desiccator together with 3 kg of dry silica gel, and the desiccator was placed in a laboratory where the temperature was kept at a constant 25 °C. The seeds were evenly buried in the silica gel, covered and sealed, and were weighed every 12 h. To extract the embryonic axis, The seed coat and the bottom ½ cotyledon were removed, and the embryonic axis, containing a small piece of cotyledon tissue, was excised with a scalpel and weighed to the same size and quality (1.5–2.5 g). The embryonic axis was then placed on an open culture dish (diameter 5 cm), and weighed at 3 h intervals. The silica gel was replaced with dry silica gel every 48 h, and the used silica gel was dried in an oven (105 °C, 2 h) before being re-used.

2.2.3. Germination Experiments of Seeds (Embryonic Axes) with Different Moisture Contents

*Q. variabilis* seeds with moisture contents of 30%, 25%, 20%, 15%, and 10% were removed from the desiccator. The seeds were placed in a glass culture dish with filter paper on the bottom, and were incubated for 30 days at 24 °C with 12 h of light and 12 h of darkness each day. The germination state of the seeds was recorded each day. The seeds were considered to have germinated when the radicle protruded more than 2 mm from the seed coat, and 15 days following germination, the seeds were transferred to a germination box to continue to germinate. Three repetitions were performed with 20 seeds per set. Seeds with initial moisture contents were used as controls. The final germination percentage, germinability, mean germination time, root length and number of rotten seeds were calculated. The calculation formulae used were as follows:

Germination percentage = (Number of germinated seeds/Number of tested seeds) × 100%

$$\text{Germinability} = \sum \frac{n_g}{t_g}$$

$$\text{Mean germination time} = \frac{\sum n_g \times t_g}{\sum n_g}$$

where $t_g$ means $g$ days after germination, and $n_g$ means the number of germinated seeds at $g$ days after germination.

For the embryonic axis germination experiments, the desiccated embryonic axes were disinfected in 75% ($v/v$) alcohol for 30 seconds (s), the alcohol was drained away, and the embryonic axes were then rinsed three times in sterile water. Embryonic axes were then disinfected for four minutes (min) in a solution of 5% sodium hypochlorite, and then rinsed three times with sterile water. The portion of the embryonic axis surface that had been saturated with the disinfectant solution was then excised, and the remaining portion was inoculated onto MJ2 germination medium (WPM basic medium, 1.50 mg/L 6-BA, 0.02 mg/L NAA, 0.50 g/L PVP, 0.30 g/L calcium nitrate, 20.00 g/L sucrose and 6.00 g/L agar, pH = 5.80).

*2.3. Thermograms of Seeds (Embryonic Axes)*

The thermograms of the cotyledons and embryonic axis of seeds with different moisture contents were investigated. The DSC determination method used here is based on the method described by Han [7] for the *C. mollissima* embryonic axis and uses D250 series (TA Company, New Castle, DE, USA) equipment. Embryonic axes and cotyledon samples from *Q. variabilis* were sealed in an aluminum pan, weighed on a microbalance, and then

cooled with liquid nitrogen ($LN_2$). The DSC analysis conditions were as follows: 30 °C for 1 min, cooling to −150 °C at 10 °C/min and then maintaining −150 °C for 2 min, and then heating to 30 °C at 10 °C/min. Three repetitions were conducted with five seeds per set.

The experimental method for determining the critical moisture content referred to and improved on that of Han [7]. The moisture content of the embryonic axis was taken as the X-axis, and the average enthalpy corresponding to the embryonic axis and cotyledon during the warming and cooling stage was taken as the Y-axis when plotting the graph. The critical moisture content is the point on the X-axis where the trend line crosses.

### 2.4. Effect of Desiccation on the Activity of Antioxidant Enzymes in Seed Cells

Referring to the method of Spitz et al. [15], micro method kits (Beijing Solarbio Science & Technology Co., Ltd., Beijing, China) were used to determine the activities of SOD, POD, MDA, and PRO in *Q. variabilis* seeds with different moisture contents. Three repetitions were conducted with five seeds per set.

### 2.5. Low-Temperature Storage of Seeds (Embryonic Axes)
2.5.1. Low-Temperature Storage of Seeds

*Q. variabilis* seeds with different moisture contents were prepared as described in Section 2.2.2 above. Seeds were then placed into 16 × 23 cm sealed bags and stored for three weeks under low-temperature conditions (4 °C, −20 °C, −40 °C, or −80 °C). Three repetitions were conducted for each experiment with 20 seeds per set.

2.5.2. Germination of Seeds at Low-Temperature Storage

*Q. variabilis* seeds with different moisture contents were removed from storage at 4 °C, −20 °C, −40 °C, or −80 °C, and were immediately thawed in a 40 °C for 3 min. Germination tests were then conducted as described in Section 2.2.3, above. Three repetitions were conducted with 20 seeds per set.

2.5.3. Selection of Embryonic Axes of Different Sizes

*Q. variabilis* seeds with the same size and weight (4 ± 1 g) were selected, the seed coats were removed and a scalpel was then used to cut off $\frac{1}{2}$ cotyledon from the bottom of the seed. The remaining portion of the seed was then cut to an embryonic axis of one of three sizes: 0.5 g, 1.5 g, or 2.5 g. Germination tests were then carried out on the embryonic axes of three sizes following the method described in Section 2.2.3. In the embryonic axes weighing 0.5 g, the only remaining tissue was the axis. The formula for the medium is given in MJ2, and the germination percentage was assessed after 2 weeks. Three repetitions were conducted with 20 seeds per set.

2.5.4. Cryopreservation of Embryonic Axes

For the cryopreservation of the embryonic axes in $LN_2$, we followed and enhanced the approach described by Lin et al. [16] for storing plant tissues in $LN_2$. The embryonic axes were placed into 2 mL cryogenic vials after being dried to varying moisture levels between 10%–30%. The method was as follows:

1.  Load procedure: the pre-chilled upload solution was added to the 2 mL cryogenic vials, and treated for 20–30 min in an ice bath. Loading solution formula: 2 M glycerol + 0.4 M sucrose, dissolved in WPM medium and adjusted to a pH of 5.8.
2.  Vitrification procedure: the loading solution was sucked up with a pipette. The vitrification solution PVS2 was pre-chilled on ice, and then added to the samples, and the samples were incubated in an ice bath for 15 min, 30 min, 45 min, or 60 min. PVS2 solution formula: 30% glycerol + 15% ethylene glycol + 15% dimethyl sulfoxide + 0.4 mol/L sucrose, dissolved in WPM medium and adjusted to a pH of 5.8.
3.  Cryopreservation: fresh PVS2 solution was replaced to the cryogenic vials, and quickly immersed in $LN_2$ for 24 h.

4.　Thawing procedure: cryogenic vials were removed from the LN$_2$, and quickly thawed at 40 °C for 3–5 min.

5.　Unloading procedure: the PVS2 solution was sucked up with a pipette, the unloading solution was added, and the samples were incubated at 25 °C for 20–30 min. The unloading solution was changed every 10 min. After incubation, the unloading solution was poured away, and the samples were shaken gently to remove as much unloading solution as possible. WPM medium was added after the unloading was finished, to remove any remaining tissue from the solution. Unloading solution formula: 1.2 M sucrose, dissolved in WPM medium and adjusted to a pH of 5.8.

6.　Germination: the method followed that described in Section 2.2.3 for embryonic axis germination. Three repetitions were conducted, with 20 seeds per set.

### 2.5.5. Thermograms of Embryonic Axis before and after Cryopreservation

The thermograms of *Q. variabilis* embryonic axes with different moisture contents were assessed both before and after cryopreservation according to the methods described in Section 2.3.

### *2.6. Statistical Analysis*

The data were organized and analyzed using Microsoft Office 2016 (Microsoft, Redmond, WA, USA). The experimental data were subjected to one-way *ANOVA* using SPSS 26.0 (IBM, Armonk, NY, USA), and when the results of the analysis of variance were considered to be significant ($p < 0.05$), multiple comparisons were made using the *Duncan*'s Multiple Range Test. The thermodynamic curve data were examined using the TRIOS V4 software (TA, New Castle, DE, USA), and the figures were prepared using the Origin 2022 software (OriginLab, Northampton, MA, USA).

## 3. Results

### *3.1. Desiccation Sensitivity of Q. variabilis Seeds*

We analyzed the germination percentage of *Q. variabilis* seeds with varying levels of moisture content. Our findings showed that the untreated seeds had a germination percentage of 95.6% (Table 1). However, as the level of desiccation increased, the seed germination percentage gradually decreased. The regression analysis shows that $R^2 = 0.941$ (Figure 1), it shows that the decrease in seed moisture content is significantly positively correlated with the decrease in germination percentage. When the moisture content was only slightly reduced (to 25%), the germination percentage remained above 75%, and desiccation had little impact on it. However, when desiccation was severe (moisture content 10%), the germination percentage of the seeds dropped to 6.30%, and the majority of the seeds lost their ability to germinate. Furthermore, when desiccation approached a moisture content of 10%, the number of seeds that rotted rose dramatically. In comparison to seeds with a natural (initial) moisture content, seeds with 30% moisture content had shorter germination times and a minor increase in root length, which suggests an improvement of seed germination ability by slight desiccation within the range of seed desiccation tolerance.

**Table 1.** Germination indicators of *Q. variabilis* seeds with different moisture content.

| Moisture Content (%) | Germination Percentage (%) | Germinability | Mean Germination Time (d) | Root Length (mm) | Number of Rotten Seeds |
|---|---|---|---|---|---|
| Initial | 95.60 ± 0.02 c | 2.21 ± 0.08 b | 8.50 ± 0.79 a | 7.78 ± 0.99 a | 0.66 ± 0.33 a |
| 30 | 93.90 ± 0.00 c | 3.47 ± 0.32 c | 7.30 ± 0.79 a | 7.82 ± 1.35 a | 0.66 ± 0.33 a |
| 25 | 77.80 ± 0.04 c | 2.05 ± 0.16 b | 11.13 ± 0.53 a | 7.02 ± 1.47 a | 1.66 ± 0.33 ab |
| 20 | 39.80 ± 0.09 b | 0.63 ± 0.05 a | 11.34 ± 2.28 a | 6.58 ± 1.24 a | 2.33 ± 0.33 b |
| 15 | 13.50 ± 0.08 a | 0.21 ± 0.10 a | 18.63 ± 1.96 b | 3.37 ± 0.83 a | 4.33 ± 0.66 c |
| 10 | 6.30 ± 0.03 a | 0.29 ± 0.25 a | 9.75 ± 4.25 a | 2.33 ± 0.44 a | 7.33 ± 0.33 d |

One-way *ANOVAs* were used to calculate the data in the table. Different letters indicated significant differences by the *Duncan* method ($p < 0.05$).

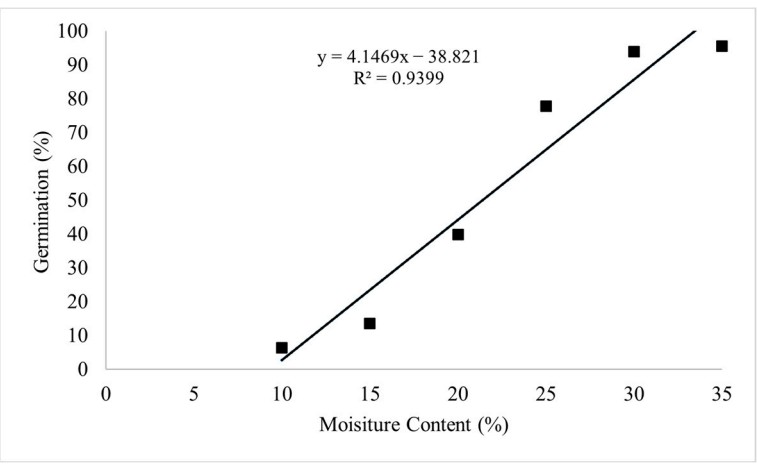

**Figure 1.** Regression analysis of germination percentage and moisture content.

The germination of *Q. variabilis* embryonic axes with different moisture contents was also assessed (Figure 2). When the moisture content was reduced from 40.83% to 20%, the germination percentage of the embryonic axes did not decline significantly, and remained over 80%. However, the germination percentage of the embryonic axes drastically decreased when the moisture content fell to 15%, and dropped to 60% when the moisture content fell to 10%. Desiccation therefore had less of an impact on the germination of the embryonic axis than on that of seeds.

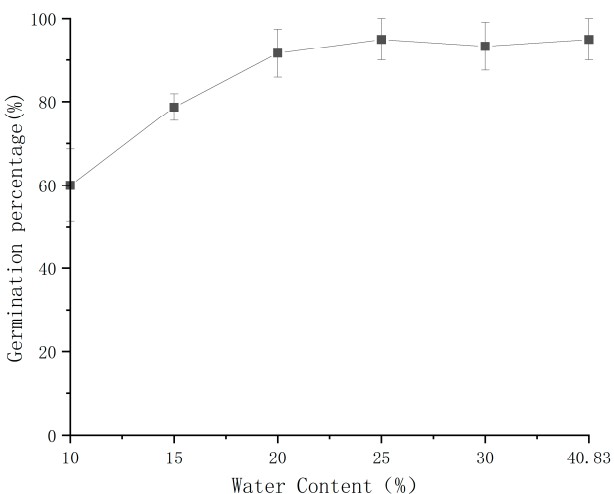

**Figure 2.** Germination percentage of Q. variabilis embryonic axes with different moisture contents.

### 3.2. Analysis of Thermograms and Determination of Critical Moisture Content

According to the thermograms of *Q. variabilis* embryonic axes and cotyledons (Figure 3, Table 2), as the moisture content of seeds decreases, the crystallization initiation temperature, peak temperature, and enthalpy also showed a regular decreasing trend. When the moisture content of the embryonic axes was reduced to 10% or 15%, the cotyledon showed no crystallization peak, although the embryonic axis had a peak. We found that the embryonic axis enthalpy values were higher than those of the cotyledon enthalpy, which indicated that the free moisture content of the embryonic axes was higher than that of the cotyledons. It is clear that the seeds of *Q. variabilis* gradually lose moisture from the outside to the inside (Table 3).

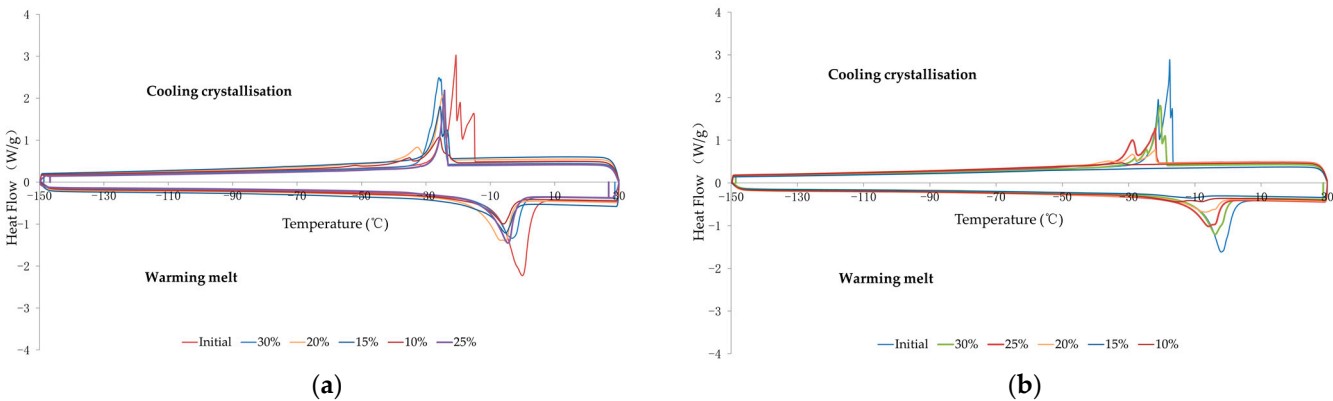

**Figure 3.** Thermograms of *Q. variabilis* embryonic axes (**a**) and cotyledons (**b**) with different moisture contents.

**Table 2.** The thermal properties of *Q. variabilis* embryonic axes and cotyledons at different moisture contents.

| Seed Site | Moisture Content (%) | Cooling Crystallization | | | Warming Melt | | |
|---|---|---|---|---|---|---|---|
| | | Onset Temperature (°C) | Peak Temperature (°C) | Enthalpy (Jg-1dw) | Onset Temperature (°C) | Peak Temperature (°C) | Enthalpy (Jg-1dw) |
| embryonic axis | 10 | −39.53 ± 6.55 a | −43.88 ± 2.67 a | 70.33 ± 6.88 a | −103.79 ± 44.89 a | −61.35 ± 44.25 a | 29.17 ± 28.20 a |
| | 15 | −34.01 ± 5.84 a | −43.25 ± 3.06 a | 76.04 ± 6.44 a | −80.25 ± 39.46 ab | −41.86 ± 33.64 a | 42.22 ± 16.40 a |
| | 20 | −32.40 ± 2.51 a | −42.55 ± 3.37 a | 94.22 ± 7.73 a | −18.22 ± 0.93 b | −9.39 ± 0.47 a | 77.73 ± 8.89 a |
| | 25 | −31.92 ± 4.02 b | −34.94 ± 5.71 ab | 125.20 ± 14.39 a | −14.69 ± 2.41 b | −6.38 ± 1.77 a | 59.54 ± 10.98 a |
| | 30 | −26.29 ± 0.94 b | −28.07 ± 2.24 b | 138.96 ± 3.72 ab | −12.55 ± 2.06 b | −4.96 ± 1.28 a | 71.88 ± 3.72 a |
| | Initial | −18.01 ± 1.73 c | −16.77 ± 1.64 c | 346.80 ± 13.73 b | −6.03 ± 0.24 b | −0.86 ± 0.19 a | 137.53 ± 6.90 b |
| cotyledon | 10 | no peak | no peak | no peak | no peak | no peak | no peak |
| | 15 | no peak | no peak | no peak | no peak | no peak | no peak |
| | 20 | −26.73 ± 2.80 a | −28.66 ± 4.06 a | 74.35 ± 3.94 a | −13.38 ± 1.93 a | −6.19 ± 0.98 a | 43.63 ± 4.43 a |
| | 25 | −21.98 ± 1.23 ab | −24.19 ± 2.81 ab | 106.67 ± 12.61 a | −9.71 ± 0.43 b | −3.88 ± 0.45 b | 55.41 ± 13.84 a |
| | 30 | −20.82 ± 1.68 b | −20.70 ± 1.90 ab | 136.67 ± 10.60 a | −9.43 ± 0.68 b | −3.78 ± 0.55 b | 55.41 ± 13.89 a |
| | Initial | −18.71 ± 1.05 b | −18.13 ± 1.12 b | 179.89 ± 40.62 a | −7.00 ± 0.51 b | −2.45 ± 0.44 b | 76.46 ± 5.44 a |

One-way *ANOVAs* were used to calculate the data in the table. Different letters indicated significant differences by the Duncan method ($p < 0.05$).

**Table 3.** Comparison of the mean enthalpy of *Q. variabilis* embryonic axes and cotyledons.

| Moisture Content (%) | Embryonic Axis Enthalpy (Jg-1dw) | | | Cotyledon Enthalpy (Jg-1dw) | | | Average Enthalpy of Cotyledons and Embryonic Axes (Jg-1dw) |
|---|---|---|---|---|---|---|---|
| | Cooling Crystallization | Warming Melt | Average Enthalpy (Jg-1dw) | Cooling Crystallization | Warming Melt | Average Enthalpy (Jg-1dw) | |
| 10 | 70.33 | 29.17 | 49.75 | - | - | - | - |
| 15 | 76.04 | 42.22 | 59.13 | - | - | - | - |
| 20 | 94.22 | 77.73 | 85.98 | 74.25 | 43.63 | 78.99 | 84.24 |
| 25 | 125.20 | 59.54 | 92.37 | 106.67 | 55.41 | 91.04 | 115.94 |
| 30 | 138.96 | 71.88 | 105.42 | 136.67 | 55.41 | 96.04 | 137.82 |
| Initial | 346.80 | 137.53 | 242.17 | 179.89 | 76.46 | 128.18 | 263.35 |

The germination percentage of *Q. variabilis* embryonic axes (Figure 2) dropped significantly between moisture contents of 10% and 20%, indicating that the critical moisture content of the *Q. variabilis* embryonic axis lies in this range. According to the linear relationship between the two (Figure 4), the calculated theoretical critical moisture content of the *Q. variabilis* embryonic axis is 11.55%

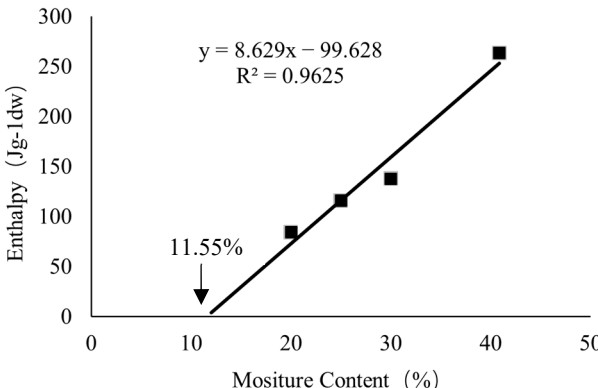

**Figure 4.** *Q. variabilis* embryonic axis moisture content vs. enthalpy.

### 3.3. Effects of Desiccation on the Antioxidant Enzyme Activity in Seeds

Reactive oxygen species are created in considerable quantities when recalcitrant seeds suffer desiccation damage, which causes lipid peroxidation of the cell membrane and the breakdown of cell structure and function [15]. In *Q. variabilis* seeds with different moisture contents (Figure 5), SOD activity and PRO activity both initially increased, then declined as the moisture content decreased further. However, POD and MDA levels increased persistently, indicating that a series of reactions were taking place in the seed cells to eliminate ROS during the desiccation process. Therefore, as desiccation increases, the seeds' stress resistance has improved and their ability to respond quickly to oxidative stress will help them minimize damage to their membrane systems, but when dehydration causes irreversible damage to seeds, the antioxidant capacity gradually decreases, and most cells die due to adversity stress.

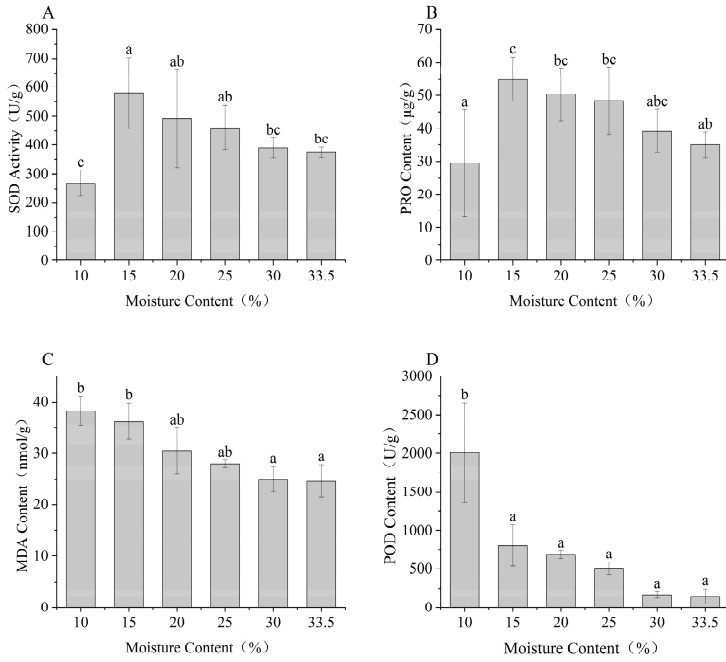

**Figure 5.** Changes in SOD (**A**), POD (**B**), MDA (**C**), and PRO (**D**) activities during desiccation of *Q. variabilis* seeds. Different lowercase letters indicate significant differences ($p < 0.05$).

### 3.4. Research on Low-Temperature Storage of Q. variabilis Seeds (Embryonic Axes)

3.4.1. Low-Temperature Storage of Seeds

The germination percentages of *Q. variabilis* seeds with different moisture contents were tested following treatment at 4 °C (Table 4). Some germination was observed in all treatment groups. After being kept at 4 °C for 21 days, the germination percentage of

*Q. variabilis* seeds stored at 4 °C did not change compared to seeds with the initial moisture content. However, the germination percentage observed in seeds in other moisture content treatment groups decreased to varying degrees. Seeds with 30%, 25%, and 20% moisture showed decreases in germination of 34.24%, 31.20%, and 13.17%, respectively, and the magnitudes of change were greater than 7.2% and 1.3% in the 15% and 10% groups, respectively, compared with the group with the initial moisture content (Tables 1 and 4). This suggests that seeds with a high original moisture content that have been desiccated are more susceptible to harm at low temperatures than seeds with a low original moisture content. In conclusion, the free moisture in seeds with a high moisture content can easily crystallize inside them at low temperatures, which damages the cells and causes the seeds to die. The vigor of these recalcitrant seeds is significantly decreased following low-temperature and desiccation treatments, demonstrating that recalcitrant seeds are sensitive to desiccation and low temperature.

**Table 4.** Germination percentage of *Q. variabilis* seeds after being stored at different temperatures for 21 days.

| Temperature (°C) | Moisture Content (%) | | | | | |
|---|---|---|---|---|---|---|
| | Initial | 30% | 25% | 20% | 15% | 10% |
| 4 °C | 94.13 ± 5.86 a | 59.66 ± 11.49 a | 46.60 ± 6.41 b | 26.63 ± 6.21 bc | 6.33 ± 0.36 c | 5.00 ± 2.88 d |
| −20 °C | 0 | 0 | 0 | 0 | 0 | 0 |
| −40 °C | 0 | 0 | 0 | 0 | 0 | 0 |
| −80 °C | 0 | 0 | 0 | 0 | 0 | 0 |

One-way *ANOVAs* were used to calculate the data in the table. Different letters indicated significant differences by the *Duncan* method ($p < 0.05$).

3.4.2. Cryopreservation of *Q. variabilis* Embryonic Axes
Germination of Different Volumes of Embryonic Axes

The germination percentage of the embryonic axes of the three different sizes were all above 85% (Table 5). There was no significant difference in the germination percentage between these groups, but the difference in percentage of contamination was significant. The smallest embryonic axes had the lowest contamination percentage, at 1.78%, while the largest and middle-sized axes had higher contamination percentage, 3.44% and 1.78%, respectively. This finding suggests that the number of cotyledons increases the susceptibility of the embryonic axis to infection. Moreover, the cotyledons are more likely to sustain irreparable damage in $LN_2$; therefore, this paper picks the smallest group of embryonic axes as the most suitable material for cryopreservation.

**Table 5.** Germination and contamination percentage of different sizes of *Q. variabilis* embryonic axes.

| Embryonic Axis Size | Germination Percentage (%) | Contamination Percentage (%) |
|---|---|---|
| Large (2.5–3 g) | 88.78 ± 2.11 a | 53.33 ± 7.26 a |
| Medium (1.5–2 g) | 88.89 ± 1.11 a | 48.33 ± 3.33 a |
| Small (0.5–1 g) | 92.22 ± 2.22 a | 1.78 ± 1.66 b |

One-way ANOVAs were used to calculate the data in the table. Different letters indicated significant differences by the Dunn method ($p < 0.05$).

The Effect of PVS2 Treatment Time on the Survival Percentage of the *Q. variabilis* Embryonic Axis following Cryopreservation

The survival percentage of *Q. variabilis* embryonic axes following cryopreservation was significantly different for embryonic axes with different moisture contents after processing in vitrification solution (Table 6). After 60 min of processing in PVS2, the survival percentage of embryonic axes with a moisture content of 10% was 15%, proving that the cryopreservation of the *Q. variabilis* embryonic axis using $LN_2$ is possible, although to date, only callus tissue was produced on germination of the axes. The experimental

method therefore requires further optimization to improve the survival percentage of the embryonic axes.

**Table 6.** Survival percentage of *Q. variabilis* embryonic axes with different moisture contents following different PVS2 treatment times after cryopreservation.

| Moisture Content (%) | Initial Germination Percentage (%) | PVS2 Processing Time (min) | | | |
|---|---|---|---|---|---|
| | | 15 | 30 | 45 | 60 |
| 10 | 60.00 ± 5.00 a | 0.00 ± 0.00 b | 0.00 ± 0.00 b | 0.00 ± 0.00 b | 15.00 ± 0.05 a |
| 15 | 78.60 ± 1.84 b | 0.00 ± 0.00 b | 0.00 ± 0.00 b | 0.00 ± 0.00 b | 0.00 ± 0.00 b |
| 20 | 91.67 ± 3.33 c | 0.00 ± 0.00 b | 0.00 ± 0.00 b | 0.00 ± 0.00 b | 0.00 ± 0.00 b |
| 25 | 95.00 ± 2.89 c | 0.00 ± 0.00 b | 0.00 ± 0.00 b | 0.00 ± 0.00 b | 0.00 ± 0.00 b |
| 30 | 93.33 ± 3.33 c | 0.00 ± 0.00 b | 0.00 ± 0.00 b | 0.00 ± 0.00 b | 0.00 ± 0.00 b |
| Initial | 95.00 ± 2.89 c | 0.00 ± 0.00 b | 0.00 ± 0.00 b | 0.00 ± 0.00 b | 0.00 ± 0.00 b |

One-way *ANOVAs* were used to calculate the data in the table. Different letters indicated significant differences by the *Duncan* method ($p < 0.05$).

### 3.4.2.3. Effect of Cryopreservation on Thermograms

During the cryopreservation of the *Q. variabilis* embryonic axes, significant changes in the moisture status of the cells may have taken place. According to the DSC measurements, embryonic axes with moisture contents of 10%, 15%, and 20% did not form crystallization peaks before cryopreservation (Figure 6), demonstrating the absence of free moisture in the cells at this stage. After cryopreservation, the 10% and 20% moisture content groups regained the crystallization peak, and free water may be recovered during the loading and vitrification stages prior to LN$_2$ treatment.

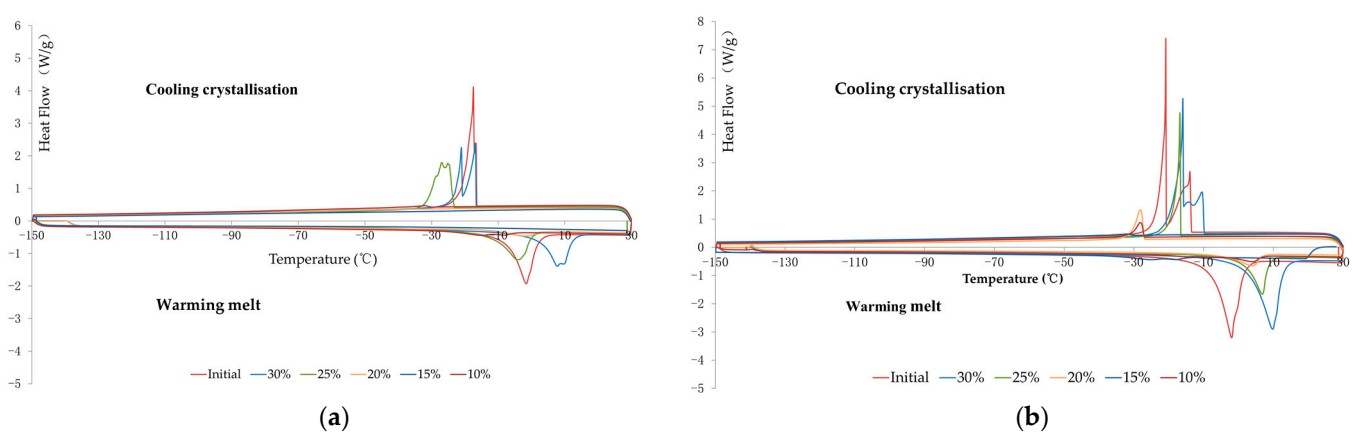

**Figure 6.** The thermal properties of *Q. variabilis* embryonic axes before (**a**) and after (**b**) cryopreservation.

## 4. Discussion

### 4.1. Desiccation Sensitivity and Critical Moisture Content

We found that *Q. variabilis* seeds were sensitive to desiccation, and can therefore be categorized as typical recalcitrant seeds. This categorization was based on the observation that the germination percentage of seeds decreased significantly as the moisture content decreased [17,18]. However, the *Q. variabilis* embryonic axis exhibited a high germination percentage (60%) even after being desiccated to 10%. This phenomenon can be attributed to two main factors. Firstly, the embryonic axis displayed a higher desiccation tolerance than the seeds. This finding is supported by the results from studies investigating germination in *Litchi chinensis*, *Euphoria longan*, and *Artocarpus heterophyllus*, where, in all cases, the embryonic axes demonstrated stronger desiccation endurance and low-temperature tolerance than their respective seeds [19]. Secondly, the difference in initial moisture content between the seed and the embryonic axis, as well as the variable desiccation rate and order, played a role. For instance, Han et al. [20] found that *C. mollissima* seeds and embryonic axes had

different initial moisture contents, with the latter having a higher initial moisture content. The DSC analysis in our study showed that the cotyledon had lower enthalpy and moisture content compared to the embryonic axis. During the desiccation process, the cotyledon desiccated first, followed by the embryonic axis. Differential rates of desiccation led to declines in cotyledon cell activity or cotyledon death, meaning that the seed lost its food source and would face challenges even under normal germination.

For recalcitrant seeds, it is crucial to examine the variations in seed moisture content carefully, as desiccation to a moisture content higher than the critical content typically has little effect on the germination of the preserved seed. In this study, we predicted that the theoretical critical moisture content of the *Q. variabilis* embryonic axis was 11.55% when combining thermodynamic characteristics with moisture content. Based on their research into seed germination and desiccation, Li et al. [21] hypothesized that the critical moisture content of *Q. variabilis* seeds was approximately 28.20%, which differed from that predicted by this study. This is likely to be as a result of the difference in the determination of seed parts and the methods of calculation. Although Li et al. [21] calculated seed moisture content and germination percentage, they calculated them based on the complete seeds. However, the moisture content and desiccation tolerance of *Q. variabilis* seeds is different from that of the embryonic axes (Table 1, Figure 2). Therefore, this study measured the desiccation sensitivity and changes in germination in both seeds and embryonic axes, and calculated the critical moisture content of the embryonic axis, which is more instructive for the cryopreservation of recalcitrant seed.

Previous research has suggested that the critical moisture content and moisture content of recalcitrant seeds are both higher than those in orthodox seeds. For instance, the initial moisture content of *Garcinia paucinervis* seeds is 45.29%, and the critical moisture content is 27.29%, while the critical moisture content of the embryonic axis of *C. mollissima* ("Yanquan") was found to be 23.90% by Han Biao et al. [20]. Feng et al. [22] found that the critical moisture content of *Ginkgo biloba* seeds ranged from 40.10% to 45.00%. Combined with the results of this study, it is clear that the critical moisture content of seeds varies from species to species. Therefore, in order to assess the critical moisture content of recalcitrant seeds and ensure the correct moisture content of seeds during storage, it is important to conduct critical moisture content determination and desiccation sensitivity studies for each species, to ensure seed viability at the time of seed preservation, and to achieve the long-term preservation of recalcitrant seeds.

How to dry the seeds to above the critical moisture content without harming them is a crucial problem, and further research is necessary in order to understand how recalcitrant seeds are sensitive to desiccation. The silica gel desiccation method used in this study has been applied in numerous studies investigating seed desiccation in different species, including *C. mollissima* [7,11], *Taxillus sutchuenensis* [23], and *Coptis chinensis* [24]. However, the desiccation of *Coptis chinensis* seeds decreased their vitality. The rapid desiccation of the seeds of certain species, including those of *Panax notoginseng* [25], was found to increase seed vitality, while delaying desiccation was advantageous for the development of desiccation tolerance in other species, such as *Pericopsis elata* [26]. We found that a portion of the seed coat eventually dried up and cracked, revealing the inner cotyledon during the rapid desiccation process when the seeds were buried in silica gel. The moisture content of the cracked, desiccated, imperfect seeds was higher than in the intact seeds with the same moisture. The authors speculate that in this case, seed desiccation occurs unevenly because of the cracks, causing the seed coat to lose moisture more quickly, while the inner cotyledons lose moisture more slowly or not at all.

### 4.2. Low-Temperature Preservation Technology

The thermodynamic properties of *Q. variabilis* seeds following infiltration treatment with PEG-6000 have been previously studied using DSC [27]. The crystallization initiation temperature in seeds with various moisture contents fell within the range of 0 °C to −40 °C. In our study, we investigated temperatures of 4 °C (above the crystallization initiation

temperature), $-20\ °C$, $-40\ °C$ (crystallization peak present), and $-80\ °C$ (below the crystallization stage). Following 21 days of storage at 4 $°C$, the germination vitality of seeds with the initial moisture content remained stable, while most other groups experienced a significant drop in germination percentage. This may be attributed to the combined damage from dryness and low temperature in the seeds. Recalcitrant seeds are vulnerable to two types of damage during desiccation: strict desiccation damage during extreme desiccation and moisture-mediated oxidative damage during mild desiccation [28]. Yan et al. [29] studied the effect of desiccation treatment on recalcitrant *Ligustrum obtusifolium* seeds and found that the germination percentage increased when they reached a certain moisture content. Other studies have also shown that the lethal moisture content of recalcitrant seeds decreases if the seeds are rapidly desiccated [30,31]. Therefore, we hypothesize that the slight improvement in germination percentage of seeds with low moisture content following low-temperature storage is related to the enhancement of seed desiccation tolerance. The synergistic effects of desiccation and low-temperature damage, which cause free water to crystallize inside the seeds and harm the cell structure, make germination difficult. Therefore, we suggest that 4 $°C$ can be used for the short-term storage of desiccated *Q. variabilis* seeds, although the preservation of these seeds needs further research.

Cryopreservation is considered to be a viable method for the long-term preservation of recalcitrant seeds. In this study, after treatment with PVS2 for 60 min, *Q. variabilis* embryonic axes with a 10% moisture content had a 15% survival percentage, with each preserved axis developing into a callus. Similar findings have been observed in other plants belonging to the genera *Castanea*, *Carya*, and *Quercus* [32]. The hormone ratio of the medium may be out of balance, and the low activity of the embryonic axis may contribute to the lack of seedling emergence. Li et al. [33] found that the survival and detoxification percentages of *Malus pumila* shoot tips were highest following 60 min of PVS2 treatment prior to cryopreservation. For *Dendrobium chrysanthum* seeds, the highest survival percentage following cryopreservation was observed after 45 min of pretreatment, and gradually declined after 60 min of treatment [34]. These findings, along with the results of the present study, suggest that the ideal PVS2 treatment time varies among different species. Numerous studies have shown that seed survival is significantly influenced by the concentration and formulation of PVS2. Further investigation is needed to determine its impact on the survival percentage of *Q. variabilis* embryonic axes. In addition to improving the quality of embryonic axes, the authors believe that further studies should optimize the choice of materials selected for cryopreservation, such as shoot tips or immature zygotic embryonic axes.

The moisture content of the seeds is a crucial element in the success of seed cryopreservation. Before cryopreservation, *Q. variabilis* embryonic axes with moisture contents of 10% and 20% did not exhibit any crystallization peaks, according to the results of DSC in this paper. However, after being stored in $LN_2$, crystallization peaks manifested in both these groups of embryonic axes, which may be one of the reasons for the survival of the 10% moisture content group. This would suggest that the material is more likely to survive when the moisture content is close to the critical moisture content during cryopreservation. However, other groups did not survive, which is likely to be because PVS2 did not adequately protect the embryonic axis during cryopreservation, allowing the free moisture to freeze and damage the embryonic axis cells, which also causes the death of embryonic axes with higher moisture contents.

We also found that the selection of the ideal recovery medium was critical. It is difficult to choose an appropriate medium for recovery growth when recalcitrant seeds are restored following cryopreservation because there are insufficient phenotype traits, which in turn results in survival failure. A recovery medium should be selected considering the properties of the same species or natural plant before cryopreservation in the future.

The preservation of recalcitrant seeds is an inherent issue in seed banks. The seeds from each plant species all have distinct tolerances for desiccation and low temperatures,

and the necessary storage methods therefore also vary. Increased research is needed to examine the desiccation sensitivity of various types and species of seeds. We cannot truly resolve the issue of recalcitrant seed storage without creating a cryopreservation technique that is broadly applicable.

## 5. Conclusions

In this study, we investigated the sensitivity to desiccation and critical moisture content in wild *Q. variabilis* seeds. We also explored methods of cryopreservation of embryonic axes. The results showed that the seeds of *Q. variabilis* are typical recalcitrant seeds, and the germination percentages of both embryonic axes and seeds exhibited desiccation sensitivity and declined as moisture content was reduced. Our findings revealed that as the moisture content of the seeds decreased, the crystallization initiation temperature, peak temperature, and enthalpy also decreased. Moreover, the theoretical critical moisture content of the embryonic axis was found to be 11.55% in our preliminary DSC study. The activities of the enzymes PRO and SOD initially increased and subsequently decreased as the degree of desiccation increased, while the activities of POD and MDA showed an upward trend. Meanwhile, *Q. variabilis* seeds with an initial moisture content retained their initial germination activity following storage at 4 °C for 21 days. The survival percentage of embryonic axes with 10% moisture and pretreated with PVS2 for 60 min was 15%, and the free moisture content of the 10% and 20% groups increased after cryopreservation, demonstrating the feasibility of LN$_2$ preservation.

Based on extensive academic research and the findings of this study, future research on *Q. variabilis* seed recalcitrance and ultra-low-temperature storage technology should focus on the following areas:

(1)  The exploration of precise seed drying techniques that allow seeds to be dried to the desired or critical moisture content in order to lessen the influence of variations in individual moisture content on the outcomes of the experiment.

(2)  The refinement of cryopreservation techniques and materials. Before the current method can be used to preserve the seeds of *Q. variabilis*, in particular, research into various parts of the *Q. variabilis* seed and various pretreatment techniques should be conducted to improve the viability of seeds (or embryonic axes) after cryopreservation. The relevant research should be extended to the entire *Quercus* genus.

(3)  The development of a reliable recovery culture system after cryopreservation, in addition to the need to optimize variables like medium formulation and disinfection techniques suitable for the growth of explants after cryopreservation.

**Author Contributions:** Conceptualization and Methodology, B.H. and B.-Q.T.; Writing—original draft, M.-J.Z. and B.H.; Data curation, M.-J.Z., Y.-Z.W. and Y.X.; Writing—review and editing, B.H. and Y.X.; Investigation, M.-J.Z., C.-C.C. and Y.-Z.W.; Project administration and supervision, B.H. and X.-M.X.; Funding acquisition, B.H. and B.-Q.T. All authors have read and agreed to the published version of the manuscript.

**Funding:** This research was funded by the Shandong Province Key R&D Program (Major Science and Technology Innovation Project) Project, grant number 2021LZGC02301, the Shandong Province Postdoctoral Innovation Project, grant number SDCX-ZG-202203059, and the Postdoctoral Station Recruitment Subsidy Project, grant number BSHCX202101, BSHCX202102.

**Data Availability Statement:** All data generated in this study are presented here.

**Acknowledgments:** We appreciate the facilitation provided by the National Wild Plant Germplasm Resource Center.

**Conflicts of Interest:** The authors declare no conflict of interest.

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
