# Peer review of "Desiccation Sensitivity Characteristics and Low-Temperature Storage of Recalcitrant Quercus variabilis Seed"

_forests, doi:10.3390/f14091837_

Round 1

Reviewer 1 Report

The paper has an interesting approach, but there are several gaps to be considered for publishing. Beyond the message of Error throughout the text ( Error! Reference source not found), which I believe is related to reference management software, the paper needs to promote the following considerations:

1. In the introduction, the first phrase mentioned a three-pattern from the physiological point of view; this is a misconception. In fact, water status in seeds has these three patterns. From a physiological point of view, it needs to associate these patterns with anhydrobiosis. In addition, in the introduction is expected a hypothesis with great relevance worldwide (Forests is an international journal). With this in mind, the introduction must be better contextualized. 

2. In the Material and Methods, several gaps need to be filled. For example, 'The Q. variabilis seeds were collected from wild individuals growing in the Culai 80 Mountain National Forest Park, China.' Fine, but what are the geographic coordinates? Other points: a. What is the initial moisture of the sample? That is so important for the physiological status of all experiments. b. Only 60 seeds (3 X 20 seeds) were used to perform the seed germination. ISTA mentioned 200 seeds. Is it representative of a population? c. all germination measurements are presented with nomenclature non-usual. For example, the Germination index must be changed for Germinability; another example, the Average mean time is the mean germination time. d. why the authors used only SOD, POD, MDA, and PRO as biological markers for oxygen metabolism? e. Although questionable, the thermodynamic was analyzed in a physical context and has a scientific argument to be analyzed from smooth curves, but germination (as a cumulative curve) and differences in moisture content must be analyzed as a quantitative treatment. Therefore, both aspects must be considered by regression analysis. That needs to be improved. f. As statistics must be reconsidered, any perspective in Results and Discussion should be avoided at this moment.

The technical language needs to be improved. For example, all measurements are in a non-usual nomenclature.

Reviewer 2 Report

Minor editorials are as follows: 

Line 2-3: Please be consistence and use upper case for each word in the Title 

Line 28: Please be consistence and use lowercase for each keyword

Line 55: Please remove "," after China 

Line 79: 2.1. Plant Materials

Line 375: Please delete the , 

activity and 

Line 384: Please delete the extra “it” which is used incorrectly

Line 413: Please delete and (-20°C, and -40°C)

Line 454: of the 10% 

Line 463: there are 

Line 464: A recovery medium

Line 485: of the embryonic

Reviewer 3 Report

The authors have presented a manuscript, describing the desiccation sensitivity characteristics and low temperature storage of recalcitrant seed Quercus variabilis. Following, I have included some comments to improve the manuscript.

  1. I suggest to the authors to add a new section detailing the state of the art. In this section, authors have to describe the relevant related work in which explain.

  1. The bibliographical references are not visible in the text, there must be a computer problem, solve it.

  1. Can the authors include at the end of the introduction, more details of the objectives of their study.

  1. This work presents very interesting results and practice to response of plant water status in Quercus variabilis. I think that the authors can improve the format of results demonstration. The authors can highlight better the importance of the results obtained.

  1. Conclusions. Consider extending the conclusions and adding a Future works paragraph. The summary and Conclusions, it is better to combine them in only section of Conclusions.

Finally, the review is interesting and presents considerable information on the possibilities of the production of Quercus variabilis, but authors must improve the presentation of their results and discussion. The topic in interesting, but the study lacks more details with precision, and concrete conclusion that will help farmers to improve or to change their strategies in the agriculture and the forest.
